# Pearl Millet–Groundnut Cropping Systems for the Sahel

**Nouri Maman [1],\* and Stephen C. Mason [2]**

[1]  Institut National de Recherche Agronomique du Niger (INRAN), Maradi BP 240, Niger
[2]  Department of Agronomy and Horticulture, 202 Keim Hall, University of Nebraska-Lincoln, Lincoln, NE 68583-0915, USA; smason1@unl.edu
\*  Correspondence: mamank.nouri@gmail.com; Tel.: +227-96-50-52-02

**Abstract:** A two-year study of pearl millet–groundnut cropping systems across two fertilizer levels was conducted in Maradi, Niger, in the Sahelian zone of West Africa. The objective of this study was to identify the best cropping system with and without fertilizer application. The experiment was conducted with ten treatment combinations of five pearl millet and groundnut cropping systems (pearl millet and groundnut sole crops, and three pearl millet—groundnut intercrops) and two fertilizer levels. Yields were approximately 300 kg ha$^{-1}$ higher in the 2022 high rainfall year than in 2021, but the year did not interact with cropping systems or fertilizer application. All the intercropping systems had an LER above 1.0, and land use efficiency increased by 19 to 41%. Fertilizer application increased yields in all cropping systems by 200 to 600 kg ha$^{-1}$. The government subsidy increased the value-to-cost ratio by 0.5 to 2.5 units and was required for the economic response for pearl millet sole and intercrops in 2022. The groundnut sole cropping system had the greatest economic response to fertilizer application. Based on the risky environment and multiple end uses needed by producers, the intercrop system M-G: 1:3:1 with fertilizer application is the best option to optimize pearl millet and groundnut production.

**Keywords:** intercrop; sole crop; land equivalent ratio; value-to-cost ratio

## 1. Introduction

Dryland cropping systems in Niger are produced on over 6 million ha with pearl millet (*Pennisetum glaucum* (L.) R. Br.) [1] and more than 85% intercropped with groundnut [*Arachis hypogea* (L.)], cowpea [*Vigna unguiculata* (L.) Walp], and grain sorghum [*Sorghum bicolor* (L). Moench]. The average yields are very low, with a grain yield of 349 kg ha$^{-1}$ for pearl millet and a pod yield of 509 kg ha$^{-1}$ for groundnut in 2021. Low yields are due to drought (low and erratic rainfall), inherent low soil fertility, infestations by pests and diseases, and inefficient crop management practices and cropping systems.

The most appropriate cropping system is based on the farmer's goals and existing environmental conditions. Only wealthy farmers who own large farms practice sole cropping systems of pearl millet and groundnut production in Niger. Traditional crop management systems are used by farmers for diverse reasons, including resilience to the incertitude of the region's climate, more durable agriculture production, and to efficiently match crop demands to the available growth resources and labor [2]. The most common advantage of intercropping is increased land equivalent ratio (LER; land use efficiency) [3] due to more efficient use of the available complementary growth resources using a mixture of crops of different rooting ability, canopy structure, height, and nutrient requirements by the component crops. Presently, intercropping is widely practiced by small-scale farmers throughout Niger and West Africa to meet diverse needs for food and fodder in markets and in subsistence and to increase farm income over that achieved by sole cropping. Intercropping system performance is commonly measured using the LER. LER is defined as the relative area needed in the sole cropping system to produce what has been obtained with the intercropping system [4]. The production of pearl millet/groundnut can be

significantly increased with the use of an appropriate cropping system, recommended fertilizer application rate, complementary genotypes, best sowing dates, row spacing, and plant population [5].

Early pearl millet (and grain sorghum) and groundnut intercropping research in semi-arid climates was conducted prior to 1990, with additional later research since 2000. Intercropping systems using short-season pearl millet varieties with long-season groundnut varieties exploit temporal differences [6–11], while with similar maturity, varieties explore spatial differences due to plant height, leaf area, or rooting patterns [9,11]. Early research found that sole crops yield more than intercrops, but intercropping increased the LER by 14 to 88% [11], with the highest LER in water-limiting production environments [8,9]. Yield and LER advantages were documented for leaf area and interception of solar radiation [11], increased conversion of light energy into dry matter [10], nitrogen fixation by the legume crop [9], lower leaf temperature of the legume crop due to shading by pearl millet [12], and differential peak demands on resources by the two crops [9]. Ref. [10] found the one-row pearl millet, three-row groundnut, and one-row pearl millet arrangement to produce the largest LER. Sorghum and pearl millet intercropping with groundnut has been shown to increase productivity and profit from N and P application over sole crops [13–15], but no study of the economics of fertilizer response has been published.

Research since 2000 confirms earlier research on yield, LER, and economic return [7,16–18], documents greater yields and fertilizer responses in higher rainfall years and with the use of water storage methods [14,15], and greater effects of pearl millet and grain sorghum on groundnut yield components than with pigeon pea (*Cajanus cajan* Millsp.) and castor (*Ricinus communis*) intercrops [19].

Pearl millet intercropping systems research largely involves using diverse plant populations applied to various plant arrangements based on the recommended plant populations in sole cropping systems. The most common approaches are to apply one of the following:

A replacement intercrop system [20], where the plant population of pearl millet is reduced and replaced by the plant population of groundnut, along with increased pearl millet row spacing.

An additive intercrop system [20], where the plant population of the pearl millet crop remains constant at the recommended sole crop plant population and groundnut is added. The objective of this study was to identify the best cropping system with and without fertilizer application to optimize pearl millet and groundnut yields, land use efficiency, and economics as measured by the value-to-cost ratio.

## 2. Materials and Methods

### 2.1. Experimental Site

Trials on the pearl millet–groundnut cropping systems were conducted during the 2021 and 2022 growing seasons at the INRAN/Tarna research station (13°27′33″ N, 07°6′14″ E). The soils of the site were classified as Arenosols [21] with a low level of clay (3.2%) and loam (0.4%) and a high level of sand (96.4%); 5.2 to 6.0 for soil pH, 1.0 to 7.8 g kg$^{-1}$ for organic C, 4.3 to 65.6 mg kg$^{-1}$ for Mehlich-3 P, and 35 to 92 mg kg$^{-1}$ for exchangeable K. The climate of the site is Sahelian, with an average rainfall of 600 mm between June and October.

### 2.2. Experimental Design

The experimental design was a randomized complete block design with three replications.

Treatment structure

The pearl millet–groundnut trial has ten (10) treatments, combinations of:

-   Five (5) cropping systems (Table 1) comprising $S_1$: pearl millet sole crop (MSC); $S_2$: groundnut sole crop (GSC); $S_3$: pearl millet/groundnut intercropped with one row of groundnut between two rows of pearl millet (M-G: 1:1:1); $S_4$: pearl millet/groundnut intercropped with two rows of groundnut between two rows of pearl millet (M-G:



1:2:1); S$_5$: pearl millet/groundnut intercropped with three rows of groundnut between two rows of pearl millet (M-G: 1:3:1); and

- Two fertilizer levels: F$_0$—no fertilizer application and F$_1$—100 kg/ha NPK (15-15-15) pre-plant applied and incorporated to the experimental area prior to planting, and 50 kg/ha urea side-dress applied to pearl millet at the 8-leaves stage.

For each treatment, the plot size was 5 m × 6 m (30 m$^2$).

**Table 1.** Pearl millet–groundnut spacing and plant population.

| Cropping System | Millet–Groundnut-Millet Distribution | Pearl Millet | | | | Groundnut | | | |
| | | Spacing | | Plant Population | | Spacing | | Plant Population | |
| | | Row | Intra-Row | | | Row | Intra-Row | | |
| | Row | ----m -- | | No./ha | % | ----- m ---- | | No./ha | % |
| Pearl Millet Sole (S1) | | 1.0 | 1.0 | 30,000 | 100 | | | | - |
| Groundnut Sole (S2) | | - | - | - | - | 0.5 | 0.2 | 200,000 | 100 |
| Pearl Millet—Groundnut Intercrop (S3) | 1:1:1 | 1.0 | 1.0 | 30,000 | 100 | 0.5 | 0.2 | 75,000 | 37.5 |
| Pearl Millet—Groundnut Intercrop (S4) | 1:2:1 | 1.5 | 1.0 | 21,818 | 72.7 | 0.5 | 0.2 | 109,091 | 54.5 |
| Pearl Millet—Groundnut Intercrop (S5) | 1:3:1 | 2.0 | 1.0 | 16,000 | 53.3 | 0.5 | 0.2 | 120,000 | 60.0 |

### 2.3. Crop Management and Data Collection

The land was plowed to a depth of 15 cm and disk-harrowed. The modern varieties were pearl millet 'Zatib' with 80 to 85 days to maturity and groundnut variety Samnut24 with 80 to 90 days to maturity [22]. The seeds were treated with Calthio C chlorpyrifosethyle 25%, Thirame 25%, and W.S (20 g/10 kg of seeds) for control of root and stem fungal diseases. The seeds were planted manually at a 5 cm depth. Pearl millet and groundnut were planted on the same day: 11 July 2021 and 18 June 2022. Two manual hand-hoe weedings were performed on 24 July 2021 and 16 July 2022, and the second on 8 August 2021 and 9 August 2022. Pearl millet was thinned to 3 plants/hill after the first weeding, and groundnut was planted with 2 seeds/hill.

Pearl millet and groundnut distribution, row and intra-row spacing, and final plant population are presented in Table 1.

### 2.4. Data Collection and Statistical Analysis

Harvest data was collected from 4 m of length for each of the four central rows for the calculation of crop yields. For the pearl millet–groundnut systems, pearl millet was harvested on 19 October 2021 and 8 October 2022. Groundnut was harvested on 16 October 2021 and 19 September 2022. Data were analyzed using analysis of variance (ANOVA) to determine variation in yield due to different cropping systems by year and combination across years using Statistix 10 software (Analytical Software, Tallahassee, FL, USA). Effects were considered significant at $p \leq 0.05$. When differences were significant, mean separation was undertaken using the least significant difference (LSD) at the 5% probability level.

The performances of intercropping and fertilizer application were evaluated with the Land Equivalent Ratio (LER). The LER for pearl millet–groundnut cropping systems was calculated using the following formulae:

$$LER = (Y_{pmic}/Y_{pmsc}) + (Y_{gnic}/Y_{gnsc})$$

$Y_{pmic}$ = pearl millet grain yield in intercropping system

$Y_{pmsc}$ = pearl millet grain yield in sole cropping system
$Y_{gnic}$ = groundnut pod yield in intercropping system
$Y_{gnsc}$ = groundnut pod yield in sole cropping system

The land equivalent ratio for fertilizer application is defined as the land required to produce, without any fertilizer application, what is produced by a land with fertilizer application for each cropping system. This $LER_f$ was calculated for each cropping by using the following formulae:

$$LER_f = \text{Yield with fertilizer} / \text{Yield without fertilizer}$$

A partial budget analysis was used to determine returns from pearl millet grain and groundnut pods applied as fertilizer to the different cropping systems. The value-to-cost ratio (VCR), which measures the average gain in the value of crop output per kg of fertilizer applied, was used. It is commonly used to evaluate the profit derived from fertilizer application, especially in the absence of data on full production costs. When VCR equals 1, the value of the yield increases over the control, which equals the cost of the fertilizer, and hence the farmer's labor input is not rewarded [22]. Technologies are considered likely for adoption when their VCRs are greater than 2.0 [23]. The VCR was calculated as a ratio of the value of increased crop output to the cost of fertilizer applied. It was calculated using the following equation:

$$VCR = \frac{(Yf - Yof) * Pr}{Cfa}$$

where:

Yf is the crop (pearl millet or groundnut) yield of a treatment in kg ha$^{-1}$ for plots with fertilizer;

Yof is the yield in kg ha$^{-1}$ for plots without fertilizer;

Pr is the price of one kg of product at harvest;

Cfa is the cost per ha of the fertilizers applied, and $(Yf - Yof)$ is the yield increase compared to plots without fertilizer.

During these two years, there were large fertilizer prices and harvested pearl millet grain and groundnut pod prices. Fertilizers were subsided by the Nigerien government in 2021, but not in 2022. To allow interpretation of agronomic and subsidy effects, the VCR was calculated. Available subsidized fertilizer prices for 50 kg bags were as follows: CFA 15,000 for NPK (15-15-15) and 13,500 FCFA for urea, thus 300 FCFA/kg and 270 FCFA/kg. In 2021, unsubsidized fertilizer prices were CFA 17,500 for NPK (15-15-15) and CFA 15,000 for urea, thus 350 FCFA/kg and 300 FCFA/kg. In 2022, unsubsidized fertilizer prices were CFA 25,000 for NPK (15-15-15) and CFA 240,00 for urea, thus 500 CFA/kg and 480 FCFA/kg.

## 3. Results

The two growing seasons of the experiment had divergent rainfall patterns, with 500 mm in 2021 and 709 mm in 2022, one near-average and the other above the average of 598 mm, and the last five years average of 611 mm (Table 2). Rainfall was greater every month in 2022 than in 2021, but especially more in September, when pearl millet grain fill and groundnut pod fill occurred.

**Table 2.** Rainfall (mm) at INRAN/Maradi station in 2021, 2022, and an average of the last 5 years.

|  | 2018–2022-Average | 2021 | 2022 |
|---|---|---|---|
| June | 74 | 54 | 80 |
| July | 147 | 150 | 122 |
| August | 222 | 226 | 293 |
| September | 145 | 71 | 209 |
| October | 11 | 0 | 1 |
| Total | 598 | 500 | 704 |

### 3.1. Agronomic Results

3.1.1. Grain and Pod Yields

The ANOVA for pearl millet and groundnut yields indicated that the interactions year*system*fertilizer, year*system, and year*fertilizer were not declared significant for both crops, indicating that the crops respond similarly to these factors across years. However, the system*fertilizer was declared significant at $p = 0.04$ for pearl millet grain yield and at $p = 0.05$ for groundnut pod yield.

The lack of year*fertilizer interaction effect indicated that pearl millet and groundnut responded similarly to the applied fertilizer during the high and near-average rainfall years in this study. Fertilizer application increased average pearl millet grain yields by 54% and average groundnut pod yields by 52%, similar to [8,9].

Averaged across years and fertilizer applications, pearl millet grain yield and groundnut pod yields were 29 to 47% greater for sole crops than with intercropping (Table 3). Thus, if the farmer's goal is to maximize production of either pearl millet grain for subsistence consumption or peanut pods as a cash crop, the producer would select a sole crop. If the producer is interested in producing multiple products to meet diverse end uses and to reduce risk [5], then he would logically choose to intercrop the two species.

**Table 3.** Influence of cropping system, fertilizer application, and year on pearl millet grain and groundnut pod yields.

| Cropping System | Row Distribution | Pearl Millet Grain | | | Groundnut Pods | | |
|---|---|---|---|---|---|---|---|
| | | Without Fertilizer | With Fertilizer | Mean (Across Fertilizer Rates) | Without Fertilizer | With Fertilizer | Mean (Across Fertilizer Rates) |
| | | ---------------------------------------------- kg/ha ------------------------------------ | | | | | |
| Pearl millet sole crop | | 828 Ba [†] | 1338 Aa | 1083 a | -- | -- | -- |
| Groundnut sole crop | | -- | -- | -- | 846 Ba | 1408 Aa | 1127 a |
| Pearl millet—groundnut intercrop | 1:1:1 | 602 Bb | 928 Abc | 765 b | 498 Bc | 699 Acd | 598 c |
| Pearl millet—groundnut intercrop | 1:2:1 | 514 Bc | 832 Abc | 673 bc | 560 Bbc | 824 Abc | 692 bc |
| Pearl millet—groundnut intercrop | 1:3:1 | 498 Bc | 658 Acd | 578 c | 683 Bb | 1013 Ab | 848 b |
| Mean (across cropping systems) | | 611 B | 939 A | | 647 B | 983 A | |

[†] Capital letters indicate row differences, and small letters indicate column differences at $p \leq 0.05$.

Pearl millet grain and groundnut pod yields were over 300 kg ha$^{-1}$ higher during the higher rainfall season of 2022 compared to the near-average rainfall season of 2021. This 13% increase for pearl millet grain yield and 10% for groundnut pod yield is consistent with previous studies [3,7,11].

When the number of rows and the plant population combinations (Table 1) in the intercropping systems were compared to sole cropping, the yield reductions with the intercropped pearl millet grain yield tended to decline with the increasing number of groundnut rows and plant population (Table 3), likely due to increased groundnut competition with pearl millet for water and light [7,24]. Groundnut pod yields were usually similar for 1:1:1 and 1:2:1 row and plant population combinations, but increased with the 1:3:1 combination, as previously reported by [10] for pearl millet and groundnut varieties with different maturities. The farmer's perceived need for pearl millet grain or groundnut pods would influence which row spacing/plant population combination would be best.

3.1.2. Land Equivalent Ratio/Land Use Efficiency

The performance of cropping systems was evaluated with the LER coefficients (Table 4). All the intercropping systems had a LER above 1.0, indicating the intercropping systems used land better than sole cropping, as previously reported [4,6–11]. The LERs of the different intercrop systems for pearl millet and groundnut were similar except for groundnut in

the 1:3:1 intercrop system, which was 0.15 to 0.22 greater than other intercrop systems for groundnut. Across years and fertilizer levels, the LER indicated pearl millet—groundnut intercropping increased land use efficiency with a large advantage over sole crops. Although fertilizer application increased both pearl millet grain and groundnut pod yields (Table 3), fertilizer application tended to decrease the LER for all intercrop systems (Table 4), as fertilizer application increased sole crop yields by over 500 kg ha$^{-1}$ compared with an increased yield of 160 to 412 kg ha$^{-1}$ for intercrop yields (Tables 3 and 4). Fertilizer application increased LERs for all intercropping systems over sole cropping systems. If the farmer's goal was to optimize land use efficiency, he would choose to intercrop the species and apply fertilizer.

**Table 4.** Influence of cropping system and fertilizer application on land equivalent ratio.

| Cropping System | Row Distribution | Cropping System without Fertilizer | | | Cropping System with Fertilizer | | | Fertilizer Application for Each Cropping System | | |
|---|---|---|---|---|---|---|---|---|---|---|
| | | Pearl Millet | Groundnut | Total | Pearl Millet | Groundnut | Total | Pearl Millet | Groundnut | Total |
| Pearl millet sole crop | | 1.0 | -- | 1.0 | 1.00 | -- | 1.00 | 1.62 | -- | 1.62 |
| Groundnut sole crop | | -- | 1.0 | 1.0 | -- | 1.00 | 1.00 | -- | 1.66 | 1.66 |
| Pearl millet— groundnut intercrop | 1:1:1 | 0.73 | 0.59 | 1.32 | 0.69 | 0.50 | 1.19 | 1.54 | 1.40 | 2.99 |
| Pearl millet— groundnut intercrops | 1:2:1 | 0.62 | 0.66 | 1.28 | 0.62 | 0.59 | 1.21 | 1.62 | 1.47 | 3.09 |
| Pearl millet— groundnut intercrop | 1:3:1 | 0.60 | 0.81 | 1.41 | 0.49 | 0.72 | 1.21 | 1.32 | 1.48 | 2.80 |

*3.2. Economic Analysis*

Groundnut pod prices were 30 to 50 FCFA kg$^{-1}$ greater than for pearl millet grain in both years, thus groundnut was of greater economic value than pearl millet in both years of this study. In all year and subsidy combinations, sole crop groundnut pods had the highest VCR, thus giving the greatest economic response to fertilizer application. All pearl millet–groundnut systems in 2021 had a VCR greater than 2 (Table 5), indicating that even with moderate rainfall, the return on fertilizer investment was justified. The fertilizer subsidy increased VCRs by 0.4 to 0.7 units. In 2022, without fertilizer subsides, despite the higher yields because of higher seasonal rainfall (Table 1), only the two cropping systems of groundnut sole cropping and the pearl millet–groundnut system (1:2:1) produced VCRs above 2 and warranted investment in fertilizer (Table 5). With fertilizer subsidies, all cropping systems produced VCRs greater than 2, thus justifying fertilizer application.

**Table 5.** Economic analysis for pearl millet–groundnut cropping systems with and without fertilizer subsidy.

| Cropping System | 2021 Without Fertilizer Subsidy | | | 2021 With Fertilizer Subsidy | | |
|---|---|---|---|---|---|---|
| | Fert. CFA | VYi CFA | VCR | Fert. CFA | Vyi CFA | VCR |
| S1: MSC | 50,000 | 131,755 | 2.62 | 43,500 | 131,755 | 3.03 |
| S2: GSC | 35,000 | 144,716 | 4.13 | 30,000 | 144,716 | 4.82 |
| S3: MGIC: 1:1:1 | 50,000 | 160,699 | 3.22 | 43,500 | 160,699 | 3.69 |
| S4: MGIC: 1:2:1 | 44,900 | 159,119 | 3.55 | 38,910 | 159,119 | 4.09 |
| S5: MGIC: 1:3:1 | 42,500 | 151,698 | 3.57 | 36,750 | 151,698 | 4.13 |

**Table 5.** *Cont.*

|  | 2022 without fertilizer subsidy | | | 2022 with fertilizer subsidy | | |
| --- | --- | --- | --- | --- | --- | --- |
|  | Fert. CFA | Vyi CFA | VCR | Fert. CFA | Vyi CFA | VCR |
| S1: MSC | 74,000 | 102,139 | 1.38 | 43,500 | 102,139 | 2.35 |
| S2: GSC | 50,000 | 182,500 | 3.65 | 30,000 | 182,500 | 6.08 |
| S3:MGIC: 1:1:1 | 74,000 | 106,247 | 1.44 | 43,500 | 106,247 | 2.44 |
| S4: MGIC: 1:2:1 | 65,840 | 145,849 | 2.22 | 38,910 | 145,849 | 3.75 |
| S5: MGIC: 1:3:1 | 62,000 | 112,556 | 1.82 | 36,750 | 112,556 | 3.06 |

Fert.: Cost per ha of the fertilizers applied (F CFA/ha); Vyi: Value of yield increase compared to plots without fertilizers (F CFA/ha); VCR: Value-to-cost ratio.

## 4. Discussion

### 4.1. Cropping System

Cropping system evaluation requires knowledge of farmer needs and priorities as well as agronomic and economic responses. In this study, it was clear that maximum pearl millet and groundnut yields would be a priority for benefits for some farmers, as sole crops produced the highest yields of both crops [13,14], and if economic return was the only goal, then the sole crop of higher economic value would be the best (Table 3). However, most poor farmers in the climate-risky Sahel zone have multiple end-use needs of food for consumption and economic return and the felt need to diversify to reduce production risk [5]. In addition, agronomic diversity through intercropping is scientifically desirable. Thus, evaluation of pearl millet—groundnut intercropping systems with the greatest grain and pod yields and land use efficiency would be desirable.

In this study, all pearl millet—groundnut intercropping systems studied had LERs greater than the one for sole cropping systems (Table 4), and thus were superior in terms of reducing the risk of crop failure and land use, as also found by others [3,5–11]. Based on yields (Table 3) and LERs (Table 4), the pearl millet–groundnut intercropping system of 1:3:1 was superior to other cropping systems, confirming the results of [10] for modern pearl millet and groundnut varieties.

### 4.2. Fertilizer Application

Fertilizer applications increased grain yield of pearl millet and pod yield in groundnut in all cropping systems studied by more than 50% (Table 3) as previously reported by [8–10,25–27] over a range of environmental conditions. However, fertilizer application slightly decreased LERs of intercropping systems, as the response to applied fertilizer was 100 to 400 kg ha$^{-1}$ greater in sole cropping than in intercropping systems. However, the fertilizer LER response was large for individual cropping systems, but especially for intercropping systems, it was very high, often more than doubling the LER. Agronomically speaking, fertilizer application was the best management practice for yield and LER optimization.

During the two growing seasons in this study, the availability of government subsidies for fertilizer was present in 2021 but not in 2022. Fertilizer application increased the VCR of sole crop groundnuts with and without subsidy in both years, supporting the research in [28]. The greatest fertilizer application response was for sole crop groundnuts, which should always receive fertilizer application. The fertilizer subsidy increased the VCR in all cropping systems, and with fertilizer application, the increase was greater in the high yield 2022 growing season than in 2021, but the fertilizer subsidy was necessary for economic returns from most pearl millet cropping systems.

Usually, the VCR for intercropping systems was the greatest for the treatment with the most rows and highest plant population of groundnut (i.e., MGIC 1:2:1 and MGIC 1:3:1). In 2021, the average rainfall year with the lowest yields, the fertilizer application always produced a VCR over 2.0 and was justified for farmer profit. In 2022, the high rainfall and high yield year, the subsidized fertilizer produced VCRs greater than 2.0, so fertilizer application was justified. However, when fertilizer was not subsidized, the VCR

was greater than 2.0 only for sole crop groundnut and MGIC (1:2:1). The reason for this response is not obvious. The VCR values in this study are based only on the economic value of grain and pod yields; thus, they are likely conservative as fertilizer application concurrently increases pearl millet [8,9] stover yields with grain yield. Stover from both crops is widely used as livestock feed and has considerable economic value.

## 5. Conclusions

A two-year study of pearl millet–groundnut cropping systems across two fertilizer levels was conducted in Niger. The objective of this study was to identify the best cropping system with and without fertilizer application to optimize pearl millet and groundnut yields, land use efficiency, and economics as measured by the value-to-cost ratio. Based on grain yield, producers with the goal of maximizing the grain yield of pearl millet to meet subsistence needs or groundnut to maximize economic return would plant sole crops and apply fertilizer if the cost was subsidized. However, if diversity and reducing production risks were the goal, then intercropping would be best. Since groundnut was of greater value, the intercropping system with a 1:3:1 arrangement was of the greatest economic value with and without fertilizer application. Government fertilizer subsidies affected the economic performance of the cropping system, as did the cropping system and year. Fertilizer subsidy increased the VCR in all cropping systems and with fertilizer application, with the increase greater in the high yield 2022 growing season than 2021. Sole crop groundnut should be fertilized in conditions similar to this study. If fertilizer subsidy is available, then application is merited to sole crop pearl millet and to intercropped pearl millet–groundnut. Based on this production environment and the multiple end-use needs of farmers, the intercrop M–G system of 1:3:1, usually with fertilizer application, can be considered the best option to optimize pearl millet and groundnut production with modern varieties with similar maturities.

**Author Contributions:** Conceptualization, N.M. and S.C.M.; methodology, N.M. and S.C.M.; software, N.M.; validation, N.M. and S.C.M.; formal analysis, N.M.; investigation, N.M.; resources, N.M.; data creation, N.M.; writing—original draft preparation, N.M.; writing—review and editing, S.C.M.; visualization, S.C.M.; supervision, N.M.; project administration, N.M.; funding acquisition, N.M. All authors have read and agreed to the published version of the manuscript.

**Funding:** This research received no external funding.

**Data Availability Statement:** Data are contained within the article.

**Acknowledgments:** We are grateful to the National Agriculture Research Institute of Niger (INRAN) for the facilities allowed to conduct the two years study and to the research technicians for their help with data collection.

**Conflicts of Interest:** The authors declare no conflict of interest.

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
