# Peer review of "Pearl Millet–Groundnut Cropping Systems for the Sahel"

_agronomy, doi:10.3390/agronomy13123029_

Round 1

Reviewer 1 Report

Comments and Suggestions for Authors

1. the title needs to be improved for fitting the whole paper perspectives, "optimizing" including too many works in cropping systems.

2. the abstract showed all the valuable details about this research, but the results about the intercrops of pearl millet-groundnut not very clear, please refined them.

3. too many keywords, generally 5-6 keywords.

4. the lines from 64-73, are these in one paragraph? 

5. do you have the temperature data about the two experimental years? 

6. over two years, the great difference existed in rainfall, did it effects the final yield and VCR?

7. the discussion, needs improving for a better clearly understand.

8. land use efficiency/land equivalent ratio, they are the same or not.

9. the conclusion should lay out based on the data and results, and with simple and significant sentence importantly. 

Comments on the Quality of English Language

based on my own proficiency and reading feel,moderate editing of English language required in this paper.

Author Response

We very grateful for you review and we would like to thank you for the time and suggestions for the manuscript quality improvement.

The reviews of the manuscript were reviewed in detail, and edits made as authors thought best.  Apologize that our health issues have slowed this process, but we have done are best and hope that Agronomy finds this manuscript adequate for publishing. Below we will address the specific points identified in the reviews.

Reviewer 2 Report

Comments and Suggestions for Authors

Author Response

(The authors gave the same response as above.)
